# Exercise-Induced Fatigue Impairs Change of Direction Performance and Serve Precision among Young Male Tennis Players

**DOI:** 10.3390/sports11060111

**Published:** 2023-05-29

**Authors:** Zlatan Bilić, Filip Sinković, Petar Barbaros, Dario Novak, Erika Zemkova

**Affiliations:** 1Faculty of Kinesiology, University of Zagreb, 10000 Zagreb, Croatia; filip.sinkovic@kif.unizg.hr (F.S.); petar.barbaros@kif.unizg.hr (P.B.); dario.novak@kif.unizg.hr (D.N.); 2Department of Biological and Medical Sciences, Faculty of Physical Education and Sport, Comenius University in Bratislava, 81469 Bratislava, Slovakia; erika.zemkova@uniba.sk; 3Faculty of Health Sciences, University of Ss. Cyril and Methodius in Trnava, 91701 Trnava, Slovakia

**Keywords:** physiological load, specific endurance, conditioning performance, consistency of strokes

## Abstract

This study investigates the effect of exercise-induced fatigue on change of direction performance and serve precision among young tennis players. A group of 21 players (age 12.90 ± 0.76 years), ranked among the top 50 players on the national tennis federation scale and the top 300 on the “Tennis Europe” scale, participated in the study. They underwent a standardized physiological load protocol using the “300-m running test” which consists of consecutive runs for 15 shares of 20 m (15 × 20). Its intensity was determined using the Borg Rating of Perceived Exertion (RPE) scale where subjects evaluated their level of experienced load on a scale from 0 to 10. Prior to and after the protocol, they performed a pre-planned change of direction *T*-test and serve precision test. Results showed significant increase of time in the *T*-test (from 11.75 ± 0.45 s to 12.99 ± 0.4 s, *p* = 0.00) and decrease in serve precision parameter from (6.00 ± 1.04 to 4.00 ± 1.26, *p* = 0.00) after the fatigue test protocol. The RPE increased from 5 to 9, after the fatigue protocol, indicating the desired fatigue effect. These findings indicate that exercise-induced fatigue impairs change of direction performance and serve precision among young tennis players.

## 1. Introduction

Performance in tennis is composed of several interrelated parameters that include technical, tactical, psychological, and functional abilities [1]. Each of these parameters needs to be developed through all stages of a tennis career. Tennis can be described as an activity characterized by a prolonged duration of repetitive high-intensity actions that are interrupted by standardized rest periods [2]. The break according to the rules of tennis is 25 s between each point, 90 s between games, and 120 s after the set is played. During tournaments, tennis players have one or two matches per day for consecutive days, depending on their successful performance and whether they play doubles in addition to singles matches. The duration and intensity of tennis matches is highly variable; however, it is not unusual for matches to last longer than three hours [3,4]. The average duration of a tennis match is between one and two hours, yet in some cases, a match can also last between three and six hours [1,5]. Many factors can affect the length of each match, such as the quality of both players, style of play, and weather conditions. Usually, matches played on clay surfaces last longer than those played on a hard court surface [5]. The level of serve precision reduces from −12% to −30% and velocity of the tennis ball by −4.5%, while at the same time the percentage of errors increases when the match takes two hours [6,7,8]. In tennis matches of extended duration, there is a significant decrease in velocity and precision of serve performance, as well as of baseline strokes [9,10], with a reduction of running capacity [10]. At the highest level of competition, the difference between successful and less successful tennis players is manifested in the ability to maintain a high percentage of stroke precision during intensive phases of the match. High-intensity periods that are energy-intensive can occur at any stage of the match, depending on the duration of the rallies in the point. Intense tournament schedule, a high number of official tennis matches, competing in both competitions (singles and doubles), and individual trainings require young competitors to both train and play unofficial matches consecutively day after day [11]. The side effect of all the previously mentioned factors is fatigue that both limits and affects sports performance simultaneously. Tennis players must respond with a higher level of physical conditioning, so that their performance can be at the highest level. Optimal physical conditioning can only be achieved by stimulating loads from competitive matches during training sessions [12]. The best results in the technical segments of the game, tennis players achieve by training in situational conditions, which is most often represented at the professional level of tennis. The appearance of fatigue during a tennis match can include a significant decline in motor skills, which primarily concerns change of direction performance and precision of serve [12].

Agility is considered one of the most important abilities for success in many sports, including tennis [13]. The ability to move quickly and cover the court during rallies defines a well-conditioned tennis player. It is defined as a rapid whole-body movement with change of velocity and/or direction in response to a stimulus [14]. The great impact of agility on the achievement of top sports results has been confirmed by numerous studies [14,15]. There are two, relatively independent, manifested types of agility. The first is non-reactive or pre-planned agility (CODS—change of direction speed), which is characterized by a change in the direction of movement that is already known in advance, that is, planned and the players do not need to react to a specific stimulus. The second form of agility is reactive or unplanned agility (RAG—reactive agility), which includes a cognitive component, i.e., observation and decision-making factors [14]. Both types of agility need to be developed equally to obtain optimal physical fitness.

Since the mentioned abilities are extremely important for achieving success in modern tennis, all research dealing with similar topics are very much useful as they provide answers as to the phase of fatigue at which the change of direction and precision of tennis players deteriorate. Various studies have shown a reduced quality of stroke performance in high level competitions during matches of extended duration [13,14,15]. Other research analyzed the effect of interval high-intensity trainings on serve precision performance among recreational tennis players, and the conclusion was made that fatigue caused by trainings significantly deteriorates the level of serve precision [16]. It is perceived that also among recreational players, fatigue has a negative effect on their serve performance quality. Upon measurements regarding the relationship between movement velocity and precision of movements during a specific exercise, it was determined that muscle and mental fatigue reduce movement velocity [17,18]. In case there is a reduction of movement velocity, the precision of performance also decreases [18]. Fatigue is most often measured and monitored by using objective methods such as heart frequency and blood lactate concentration, as well as subjective methods, such as assessments of load sensation by means of questionnaires or diaries. These methods are simple, inexpensive, and easily available [19]. That is why these methods for assessing fatigue can be used on the tennis court, without many distracting factors when testing is acquired. These methods, due to their simplicity, with simple instruction from professionals, can also be conducted by tennis and fitness coaches on tennis courts. There are many simple devices on the market that can measure heart rate and lactate levels, which are easily accessible to a professional team of tennis players. The effects of fatigue in tennis can be manifested as unforced errors, reduced velocity and precision, inadequate positioning (reduced velocity of footwork, and bad preparation for the stroke) and incorrect tactical choices decisions [13,20]. Since tennis players are constantly subject to fatigue during the match, the essential problem lies in the fact that there is not much research that attempted to determine the effect of fatigue on the level of motor abilities, thus in turn also on their efficiency during a tennis match. Furthermore, previous research mostly referred to senior players [6,10,11], while little is known about the extent to which intensive specific activity affects the key abilities among young tennis players. Assuming that most young tennis players are not as physically prepared as senior tennis players, precision of serve during significant stages of fatigue in young tennis players is probably lower. Because of all of the above, it is important to do testing with a sample as it is in this study.

The aim of this study was to investigate the effect of exercise-induced fatigue on change of direction performance and serve precision among young tennis players. It is expected that fatigue-induced by high intensity repeated exercise significantly reduces change of direction performance and the number of accurately executed tennis strokes.

## 2. Materials and Methods

### 2.1. Participants

The sample consisted of 21 male tennis players with the age 12.90 ± 0.76 years, height 157.61 ± 6.65 cm, and body mass 48.31 ± 5.02 kg, who are ranked among the top 50 players on the national tennis federation rankings, and among the top 300 players on the international “Tennis Europe” rankings. Of the sample, 15 participants were right-handed, 6 were left-handed, and all had two-handed backhand stroke. The G-power program was used to estimate the corresponding number of participants of 21 with the expected effect power f = 0.33 m alpha level of 0.05 and statistical power of 0.90 (version 3.1.9.4; Heinrich Heine University, Dusseldorf, Germany). The tests were conducted on outdoor clay tennis courts. Inclusion criteria included being in good health, physically active players who train at least three times per week and compete in regional, national, or international tournaments. Exclusion criteria were any injury that influences tennis play and physical performance. Each participant was informed on the topic and aim of the research, and all examinees, as well as their parents, provided written consents for participating in the study. Moreover, they were asked to refrain from high-intensity exercises to avoid the impact of fatigue and physical load on testing results. The overall testing protocol was explained in detail, with particular emphasis on the fact that the research required a certain amount of additional load and presented a risk for injury equal to the risk level during the standard training process or competition. The study was conducted in accordance with the 1964 Helsinki Declaration and its later amendments, and it was approved by the Institutional Review Board of the Faculty of Kinesiology University of Zagreb (protocol code 12; date of approval 22 February 2023).

### 2.2. Measurements

A standardized *T*-test, which includes frontal and lateral movements, such as specific tennis player moves during a match, was used to measure change of direction speed [14]. *T*-test was measured in both directions. Since the change of direction in movement during this test was known in advance, pre-planned agility was evaluated in this research. The achieved results during the test were measured by using PowerTime system photocells (Newtest Oy, Oulu, Finland). The serve precision variable was evaluated with the number of successful first serves in accordance with the rules of tennis, while each of the participants had a total of 10 serve attempts. For a serve to be successful, the player had to hit any spot in the serve box. Each participant first served 5 serves to the “deuce” serve box, whereas the following 5 serves were performed to the “advantage” serve box.

### 2.3. Experimental Protocol

Prior to conducting the testing, all participants performed a standard tennis warm-up for a period of 15–20 min. The warm-up protocol included light-intensity running over 10 lengths of 20 m, after which followed dynamic stretching exercises for a total of 15 min (lateral movements, skipping, jumping, lunges, and, finally, 4 lengths of sub-maximum acceleration). In the end, each participant, for the purpose of serve warm-up, performed 20 first serves to the “deuce” and to the “advantage” serve box. The warm-up was followed by the initial implementation of the change of direction and serve precision tests, then the fatigue test protocol, and the final change of direction and serve precision tests. Participants performed the standardized physiological load running protocol until exhaustion. The “300-m running test” consists of consecutive runs for 15 shares of 20 m (15 × 20) at maximum velocity without rest. The 300-m shuttle run test (300 m) estimates the level of anaerobic power ability and therefore belongs to the group of tests that measure functional ability of athletes. The test is performed in a sports hall or outdoors. Two 1-meter-long parallel lines are drawn 20 m apart. The athlete begins at the starting line on a signal by a timekeeper and runs 15 lengths at maximum speed ensuring that s/he steps on the line before turning around and running back to the next line. The timekeepers record the split times of every length covered as well as the final result. The “300-m run test” is a useful and valid field test of anaerobic capacity, commonly used in previous research conducted on young athletes [21,22,23]. All participants have the same conditions (temperature, humidity) during the testing.

For the assessment of intensity of the experienced load, the Borg Rating of Perceived Exertion (RPE) was used, where each participant determined the load level on the Likert scale from 0 to 10 during the performance of initial and final tests. The higher number on the scale indicated a higher level of subjective load.

The rest interval after the fatigue protocol was 30 s, as this was the time required for the participant to reach the starting position and prepare for the final performance of *T*-test. Upon completing the test, the participant immediately started with the final performance of the serve precision test.

### 2.4. Statistical Analysis

Basic descriptive parameters (mean—$\bar{x}$; standard deviation—SD) were used to describe each variable. The normality of the distribution was tested with the Kolmogorov–Smirnov test. A repeated measures MANOVA was used to assess the effect of exercise-induced fatigue on change of direction performance and serve precision. Moreover, correlation between results in change of direction and serve precision test (independent variables) with the ranking of the players (dependent variable) was tested with a multiple regression analysis test. No multicollinearity problems were confirmed. Statistical analysis was performed with the use of Statistica 14.0.1.25 (TIBCO software, Inc., Palo Alto, CA, USA). The level of statistical significance was set at *p* < 0.05.

## 3. Results

Results showed significant increase of time in the pre-planned change of direction *T*-test (from 11.75 ± 0.45 s to 12.99 ± 0.40 s, *p* = 0.00) and decrease in serve precision parameter expressed by “deuce” and “advantage” serve box (from 6.00 ± 1.04 to 4.00 ± 1.26, *p* = 0.00) after the fatigue test protocol. The RPE increased from 5 to 9, after the fatigue protocol indicating the desired fatigue effect (Table 1).

Multiple regression analysis were used to find the relationship between several independent variables (agility *T*-test-pre, RPE-pre, serve precision-pre, agility *T*-test-post, RPE-post, serve precision-post, and a dependent variable (ranking) in both group. Players were divided into two subgroups to determine whether the players with the better or worse performance present different fatigue scales. The first group (*N* = 11) was of higher ranked players (119–219 rankings on Tennis Europe) and the second group (*N* = 10) was of lower ranked players (219–289 rankings on Tennis Europe).

According to the results of regression analysis, the coefficient of multiple correlation of 0.42, and the coefficient of significance *p* < 0.79, it can be concluded that there is no statistically significant correlation between the subgroup of players ranked 219–289 on Tennis Europe rankings with tested parameters of precision of serve and agility performance (change of directions). The determination coefficient of 0.42 indicates that a whole set of independent variables share 42% of common variance (Table 2).

According to the results of regression analysis, the coefficient of multiple correlation of 0.39, and the coefficient of significance *p* < 0.88, it can be concluded that there is no statistically significant correlation between the subgroup of players ranked 119–219 on Tennis Europe rankings with tested parameters of precision of serve and agility performance (change of directions). The determination coefficient of 0.39 indicates that a whole set of independent variables share 39% of common variance (Table 3).

## 4. Discussion

Exercise-induced fatigue significantly affected the change of direction performance and serve precision during stroke performance in tennis players. Such results have both scientific and practical applicability since the mentioned abilities are essential for performance and efficiency in tennis. After the “300-m running” test, there was a significant decrease in agility time of tennis players with a result of 12.99 s compared to the initial result of 11.75 s, which shows the onset of satisfactory fatigue prior to the serve precision test implementation. A player who in the intense stages of the match, during accumulated fatigue, can be more precise in performing serves and faster in moving towards the ball and covering the court, creates a greater advantage to win the match. These findings are in accordance with previous studies [20,24], where fatigue significantly affects abilities related to speed and explosiveness, but also certain neurophysical aspects which considerably contribute to sports performance in racket sports. The decrease of agility upon the appearance of fatigue which is represented in all changes of direction during the implementation of the standardized *T*-test for assessment of ability to change direction [24]. On a sample of 292 tennis players and 146 played matches in the main draw of the US Open Grand Slam tournament, it was determined that fatigue significantly affects performance and tactical adjustment during a tennis match, and therefore also reduces the velocity in changes of direction in any way [25].

The results of this, but also of previous studies [24,25], confirm the presumption that tennis players who are subject to fatigue need more time for complete regeneration, and that therefore also their motor abilities are reduced when compared to the beginning of the training or match. The average results for serve precision prior to the fatigue protocol was six successful serves, while after the fatigue protocol the number was significantly reduced to an average of four successful serves. It is expected that tennis players, immediately after the intensive fatigue test, will have less ability to achieve the same good results as before the fatigue test in all motor skills that can be tested on the court. By training the service to make technical progress, it can certainly be a good basis for maintaining a high level of precision of service even in the intensive stages of a tennis match. It can be noticed that upon the onset of fatigue, there is a decline of serve precision; however, it is also to be expected that due to fatigue and decline in agility, there shall also be a poorer positioning in preparation for the upcoming tennis ball, which shall in turn result with inaccurate forehand and backhand strokes. On a sample of 30 young tennis players with the average age 19.5 ± 3 years, they assessed the precision of baseline tennis strokes (forehand and backhand) among higher and lower-ranked players after implementing the fatigue protocol by using the Loughborough load test. During high-intensity load (90% of maximum heart rate) in lower-ranked players, there was a significant 49.6% reduction of precision for all strokes, while in higher-ranked players their precision reduced by 40.3% [26].

On a sample of 18 tennis players with the average age of 20.07 ± 0.9 years, the effect of acute fatigue on motor abilities and physiological parameters by using the Loughborough load test reduced the precision of basic strokes by 69%, whereas serve performance decreased by 30% [13]. Fatigue and lapse of concentration considerably affect parameters of situational efficiency in serve and serve return in all Grand Slam tournaments, regardless of the tennis court surface [27,28]. Repetitive load causes fatigue, and thus in turn reduces the level of motor abilities and precision performance of all tennis strokes, while our study implemented on examinees aged 12.90 ± 0.76 years also leads to the conclusion of a significantly reduced serve precision. Such findings point to the relevance of specific endurance of tennis players and the fact that players capable of fatigue postponement for longer periods shall also achieve better results. In addition to physical fatigue, the question is raised on the level of mental fatigue as a result of load, that is, on the lapse of concentration which in turn affects the performance and precision of strokes.

The greatest difference In all the measured variables in this research was found for the rating of perceived exertion (RPE) variable where the examinees during the performance of initial tests evaluated the level of experienced load on a scale from 0 to 10 with a result of 5, whereas after the fatigue protocol and the implementation of identical tests, this number significantly increased to a result of 9, which indicated that the “300-m running” test achieved the desired fatigue effect. It is possible that the biggest difference in this variable is due to the fact that the tested players are not at a top level of fitness preparation and are not used to high-intensity actions.

The rating of perceived exertion method provides reliable and valid results, as well as representing a very good instrument for training of specific endurance in tennis [3,29]. Since specific endurance is a key factor for postponement of fatigue during the game and for maintaining the level of abilities at the required level, it is precisely trainings for young tennis players that should be targeted towards acquiring the best possible level of specific functional physical preparation. Likewise, it should be emphasized that service performance technique is at a very good level among young tennis players; however, it is certainly also a motor skill that is still under development, as well as the skill of performing other basic strokes. For the above-stated reason, fatigue can significantly affect the decrease of precision and competitive efficiency.

Based on the results obtained with multiple regression, the analyzed tests have a better prediction in the first group (players ranked 119–219) for achieving a better competitive level, i.e., ranking. (R^2^ = 0.42 vs. R^2^ = 0.39).

This research is manifested in the fact that prior studies [28,29] mostly referred to senior players, while there were little findings on how and to what extent intensive specific activity affect the key abilities and skills in young tennis players. This study had several limitations, which are discussed below. First, the subjects involved in this study were selected youth tennis players in a very sensitive and crucial developmental phase. Secondly, we did not evaluate the biological age of the participants, which is known to influence neuromuscular performance. Future research should be directed towards implementing tests with a larger number of examinees in competitive conditions in order to obtain data and thus to enable explanations for the appearance of fatigue and its effect on abilities in younger tennis players during matches. In addition, the range of landing points should be set in the service area to be like the requirements of the competition. Further, kinematic parameters affected by fatigue could be tested, muscle activity and metabolic cost as well.

Moreover, for the fatigue protocol, it would be interesting to choose a more specific and progressive on-court endurance tennis test, and for the serve precision more precise targets with smaller dimensions of the service box. Future research should include measurement of the individual fitness level of participants and heart rate measurements as well.

The gained insights from this research in practice can be for the tennis and fitness coaches to understand changes in the tested parameters during exercise-induced fatigue. Based on the results of the tested sample, they can predict by what percentage the level of precision and change of direction decreased during the fatigue phases of match on their tennis players with similar age range. This can be used in planning and programming training loads in order to improve the above parameters.

## 5. Conclusions

Exercise-induced fatigue significantly affects the changes affects the change of direction speed on the tennis court (by 11%) and the precision of serve performance (by 20%) in younger tennis players, which can result in a subordinate position during a tennis match, especially in service games. The above-stated specifies the relevance of specific physical conditioning preparation in accordance with the demands of tennis in competitive categories for U12 and U14 players. This measurement protocol can help the coaching team as a good example for testing the change of movement and precision of playing other shots in tennis.

## Figures and Tables

**Table 1 sports-11-00111-t001:** Pre-post fatigue test protocol changes in agility performance and serve precision.

Variable	Pre	Post	F	*p*	ES
Agility “*T*-test” (s)	11.75 ± 0.45	12.99 ± 0.4	84.11	0.00 *	0.78
Serve precision (1–10)	6.00 ± 1.04	4.00 ± 1.26	31.11	0.00 *	0.66
Rating of perceived exertion (1–10)	5.00 ± 0.84	9.00 ± 1.22	225.30	0.00 *	0.89

Legend: ES—effect size; *—significant interaction (*p* < 0.05).

**Table 2 sports-11-00111-t002:** Overview of multiple regression analysis for first group (119–219 rankings).

*N* = 11	R = 0.65/R² = 0.42/F(6.4) = 0.48/*p* < 0.79/Std.Error of Estimate: 32.67					
	beta	Std.Err.of b	b	Std.Err.of b	t(4)	*p*-Value
Intercept			−284	422	−0.67	0.53
Agility *T*-test-pre	0.08	0.45	6.86	36.14	0.18	0.85
RPE-Pre	0.07	0.43	2.66	15.20	0.17	0.86
Serve precision-pre	−0.11	0.49	−3.89	16.10	−0.24	0.82
Agility *T*-test-post	0.62	0.45	38.61	28.40	1.35	0.24
RPE-Post	−0.10	0.67	−3.19	20.89	−0.15	0.88
Serve precision-post	−0.06	0.66	−1.43	15.39	−0.09	0.93

**Table 3 sports-11-00111-t003:** Overview of multiple regression analysis for second group (219–289 rankings).

*N* = 10	R = 0.62/R² = 0.39/F(6,3) = 0.32/*p* < 0.88/Std.Error of Estimate: 43.95					
	beta	Std.Err.of b	b	Std.Err.of b	t(3)	*p*-Value
Intercept			−382	1301	−0.29	0.78
Agility *T*-test-pre	1.11	1.37	63.10	78.04	0.80	0.47
RPE-Pre	−0.17	1.63	−7.19	67.43	−0.10	0.92
Serve precision-pre	−0.16	1.35	−4.15	34.34	−0.12	0.91
Agility *T*-test-post	0.05	0.94	4.60	80.15	0.05	0.95
RPE-Post	−0.69	0.91	−25.97	33.98	−0.76	0.50
Serve precision-post	0.51	2.04	12.11	47.77	0.25	0.81

## Data Availability

Data available on request.

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
