# Peer review of "Exercise-Induced Fatigue Impairs Change of Direction Performance and Serve Precision among Young Male Tennis Players"

_sports, 2023, doi:10.3390/sports11060111_

Round 1

Reviewer 1 Report

The proposal paper is very suitable from a methodological point of view and surely will be at starting point for futures interventions according to the data collected.

The sample, the experimental protocol and procedures have been very well described and applied, and the statistical analysis, from my point of view, has been too correct to study the results from the research.

Concerning with one of the variables studied, the precision, I have some questions for the authors:

Accuracy is only evaluated if the ball is "inside", and not with some more precise target (smaller dimension) of the service area.

What was the criterion for selecting the sample, because for 21 players there is only one factor: National Ranking (within the top 50) and European Ranking (within the TOP 300), and this dispersion (0-300) could affect the results in terms of accuracy (level of play of the players) ?

On the other hand, a neuromuscular variable affected by fatigue could be accuracy, but also speed, which is crucial in this stroke (even more than accuracy) and, however, it has not been recorded.

Could you tell me if it was ruled out for some issue?

Finally, I consider that in addition to the limitations explained by the authors, there is another one that could affect the title of this paper, it is related to the gender of the sample, since if we only study male players, and especially at this stage (biological differences increase), perhaps we should be more specific in the title so as not to confuse the readers.

Reviewer 2 Report

Although the work seems very interesting and useful in terms of its applicability to the field, it is based entirely on the hypothesis that physiological fatigue can be induced by an induction of physical exertion related to a "300 meters" test, for which only one reference is inserted whose link does not work. Since I cannot verify the accuracy and appropriateness of including this test, I do not believe I can approve the work in this format.

Round 2

Reviewer 2 Report

As a result of carefully reading the modifications made by the authors, I have verified that to the best of my knowledge, the references linked to the 300M test are sufficient to establish that the updated point study based on this test is accurate at this point in time.

There has been a change in my perspective from my original decision, and I now consider the study acceptable for publication.